# Mutations in the F protein of the live-attenuated respiratory syncytial virus vaccine candidate ΔNS2/Δ1313/I1314L increase the stability of infectivity and content of prefusion F protein

Judith Alamares-Sapuay[1], Michael Kishko[1], Charles Lai[1], Mark Parrington[1], Simon Delagrave[1¤], Richard Herbert[2], Ashley Castens[2], Joanna Swerczek[2], Cindy Luongo[3], Lijuan Yang[3], Peter L. Collins[3], Ursula J. Buchholz[3], Linong Zhang[1]*

**1** Sanofi, Cambridge, Massachusetts, United States of America, **2** Experimental Primate Virology Section, Comparative Medicine Branch, National Institute of Allergy and Infectious Diseases, National Institutes of Health, Poolesville, Maryland, United States of America, **3** RNA Viruses Section, Laboratory of Infectious Diseases, Division of Intramural Research, National Institute of Allergy and Infectious Diseases, National Institutes of Health, Bethesda, Maryland, United States of America

¤ Current address: Delagrave Life Sciences Consulting, Sudbury, Massachusetts, United States of America
* Linong.Zhang@sanofi.com

## Abstract

Respiratory syncytial virus (RSV) is the leading viral cause of bronchiolitis and pneumonia in infants and toddlers, but there currently is no licensed pediatric vaccine. A leading vaccine candidate that has been evaluated for intranasal immunization in a recently completed phase 1/2 clinical trial is an attenuated version of RSV strain A2 called RSV/ΔNS2/Δ1313/I1314L (hereafter called ΔNS2). ΔNS2 is attenuated by deletion of the interferon antagonist NS2 gene and introduction into the L polymerase protein gene of a codon deletion (Δ1313) that confers temperature-sensitivity and is stabilized by a missense mutation (I1314L). Previously, introduction of four amino acid changes derived from a second RSV strain "line 19" (I79M, K191R, T357K, N371Y) into the F protein of strain A2 increased the stability of infectivity and the proportion of F protein in the highly immunogenic pre-fusion (pre-F) conformation. In the present study, these four "line 19" assignments were introduced into the ΔNS2 candidate, creating ΔNS2-L19F-4M. During *in vitro* growth in Vero cells, ΔNS2-L19F-4M had growth kinetics and peak titer similar to the ΔNS2 parent. ΔNS2-L19F-4M exhibited an enhanced proportion of pre-F protein, with a ratio of pre-F/total F that was 4.5- to 5.0-fold higher than that of the ΔNS2 parent. The stability of infectivity during incubation at 4˚C, 25˚C, 32˚C and 37˚C was greater for ΔNS2-L19F-4M; for example, after 28 days at 32˚C, its titer was 100-fold greater than ΔNS2. ΔNS2-L19F-4M exhibited similar levels of replication in human airway epithelial (HAE) cells as ΔNS2. The four "line 19" F mutations were genetically stable during 10 rounds of serial passage in Vero cells. In African green monkeys, ΔNS2-L19F-4M and ΔNS2 had similar growth kinetics, peak titer, and immunogenicity. These results suggest that ΔNS2-L19F-4M is an improved live attenuated vaccine

**Data Availability Statement:** All relevant data are within the manuscript and its Supporting Information files.

**Funding:** This study was funded in part by Sanofi. This research was also supported in part by the Division of Intramural Research of the NIAID, NIH. U.J.B., C. Luongo, L.Y., and P.L.C. were supported by a Collaborative Research and Development Agreement with Sanofi on the development of RSV vaccines.

**Competing interests:** J.A.-S., M.K., C. Lai., M.P., S. D. and L.Z. are current or former employees of Sanofi and may hold Sanofi stocks. C. Luongo., P. L.C., and U.J.B. are inventors on patents covering part of this material, filed by the United States, Department of Health and Human Services.

candidate whose enhanced stability may simplify its manufacture, storage and distribution, which merits further evaluation in a clinical trial in humans.

## Introduction

RSV is an enveloped, non-segmented, negative sense, single-stranded RNA virus that belongs to the *Pneumoviridae* family [1]. Its genome is approximately 15,200 nt in length and contains 10 genes in the order 3′ NS1-NS2-N-P-M-SH-G-F-M2-L [2, 3]. Each gene is transcribed into an mRNA encoding a single protein except for the M2 mRNA, which contains two overlapping open reading frames (ORFs) encoding two different proteins, called M2-1 and M2-2. The viral envelope contains 3 transmembrane surface glycoproteins: the glycoprotein (G), the fusion protein (F), and the small hydrophobic protein (SH). The matrix M protein and the above-mentioned M2-1 protein form two separate shells inside the viral envelope. The ribonucleoprotein (RNP) complex consists of the viral RNA and four proteins essential for RSV RNA synthesis: the nucleoprotein (N) that tightly binds along the entire length of the viral RNA, the large polymerase subunit (L), the phosphoprotein (P), and the M2-1 protein that forms a virion shell as noted above and also functions in the nucleocapsid/polymerase complex as a transcription processivity factor. The nonstructural proteins NS1 and NS2 interfere with innate immune responses including interferon induction and signaling [4, 5], and NS2 has a role in promoting the shedding of infected cells [6].

RSV causes a substantial disease burden in infants, as well as immunocompromised and elderly populations, with nearly all children getting infected one or more times with RSV by 2 years of age [7]. It is a leading cause of bronchiolitis and pneumonia in infants and children, and severe RSV disease in early childhood has been associated with higher rates of recurrent wheezing and asthma in children later in life [8–10]. Worldwide, RSV is estimated to cause 3.6 million hospital admissions and 85,000 to 125,000 deaths in children under 5 years of age annually [11]. Maternal antibodies offer some protection to newborn infants but require high titers to efficiently protect the respiratory tract. To protect infants against RSV illness during their first months of life, two products for passive immunoprophylaxis are available, including Palivizumab (Synagis), a costly, monthly, injection offered only to infants at the highest risk of complications [12] and the recently licensed, extended half-life monoclonal antibody Nirsevimab (Beyfortus), with seasonal administration recommended for all infants under 8 months of age [13]. Additionally, an RSV pre-F subunit vaccine for maternal immunization during pregnancy recently received regulatory approval [14]. In Phase 3 studies, both of these approaches were effective against medically attended severe RSV-associated lower respiratory tract illness in infants [13, 14]. However, maternal antibodies or antibodies from passive immunoprophylaxis will wane [15, 16]. To reduce the disease burden in older infants and toddlers, a vaccine with the ability to induce active immunity against RSV is essential.

The development of a pediatric RSV vaccine for active immunization has been hampered by the experience in the 1960s with a formalin-inactivated RSV (FI-RSV) vaccine that was evaluated in infants and young children. This vaccine was poorly protective and in RSV-naïve recipients it induced an altered immune response that caused an enhancement of RSV disease upon natural RSV infection [17–19]. Subunit RSV vaccines similarly appear to prime for enhanced disease in experimental animals [20]. In contrast, live-attenuated RSV vaccines (LAVs) are not associated with enhanced RSV disease and have been shown to be safe for RSV-naïve recipients [21, 22]. A number of attenuated versions of RSV have been developed,

most recently using reverse genetics, and are being evaluated as live vaccines for intranasal administration.

One leading candidate, ΔNS2, was derived from recombinant RSV A2 and contains two attenuating elements, namely (i) the deletion of the RSV interferon (IFN) antagonist NS2 gene, and (ii) the deletion of codon 1313 in the polymerase (L) gene which renders RSV temperature-sensitive [23], with a shutoff temperature for viral replication of 38˚C to 39˚C. The ΔNS2 virus also contains an I1314L missense mutation that had been found to stabilize the Δ1313 mutation, but is not attenuating itself [23, 24]. The ΔNS2 candidate exhibited promising levels of safety, attenuation, and immunogenicity in phase 1 studies in 6–24 month-old infants and children [24] and has been evaluated in a recently completed phase 1/2 clinical trial (NCT04491877).

The G and F glycoproteins are the two viral neutralization antigens and are the major protective antigens [25–27]. RSV infects and replicates predominantly in airway epithelial cells. *In vivo*, the G protein mediates virus attachment via a CX3C motif that binds to the CX3CR1 receptor expressed on ciliated airway epithelial cells [28–30]. *In vitro*, cultures of differentiated primary human airway epithelial (HAE) cells that express CX3CR1 on their apical surface, can be used to model RSV infection [28–30]. The F protein mediates cell fusion, resulting in viral entry and infection. When newly synthesized and cleaved, the RSV F trimer assembles in the metastable, fusion-active pre-F conformation. Upon contact with the cell surface, the F protein undergoes an irreversible conformational change that drives fusion between the viral envelope and host cell plasma membrane [31, 32]. However, this conformational change is easily triggered prematurely and results in the accumulation of non-functional F protein in the post-fusion (post-F) conformation on the virus surface and a decreased proportion of virion-bound F in the pre-F conformation, reducing infectivity.

Several studies demonstrated that the majority of F-specific neutralizing antibodies are specific to the pre-F conformation [33–35]. Therefore, increasing the proportion of F protein in the pre-F conformation would be expected to increase the immunogenicity of an RSV vaccine. Elucidation of the structure of pre-F has enabled the design of F protein mutants that have increased stability in the pre-F conformation [36–39]. However, these modifications generally render the F protein inactive for fusion and thus cannot readily be used in live-attenuated RSV vaccines. Stobart et al. [40] previously identified four amino acid differences between the F proteins of RSV strain A2 and a second strain called "line 19" (I79M, K191R, T357K and N371Y) that appeared to increase the thermal stability of infectivity as well as the proportion of F protein in the pre-F conformation without compromising viral infectivity and replication.

An increase in the stability of infectivity and the proportion of pre-F protein in a live-attenuated RSV candidate might increase the efficiency of vaccine manufacture and also increase immunogenicity. However, it would be important to determine whether this involves changes to replication, infectivity, or attenuation that might alter vaccine tolerability and performance. In the present study, we incorporated these four "line 19" mutations into the ΔNS2 vaccine candidate to generate the modified candidate ΔNS2-L19F-4M, and evaluated its properties with regard to (i) *in vitro* replication in Vero cells and HAE cultures, (ii) thermal stability, (iii) content of pre-F protein, (iv) genetic stability during replication in Vero cells, and (v) infectivity, replication, and immunogenicity in African green monkeys (AGMs).

## Materials and methods

### Cells and viruses

Vero CCL 81.2 cells were obtained from the ATCC and maintained in Dulbecco's Modified Eagle's Medium (DMEM, Sigma D6546) supplemented with 10% fetal bovine serum (FBS,

Gibco 10100–147) and 1% Glutamax (Thermo Fisher Scientific 35-050-061). Primary undifferentiated human airway epithelial (HAE) cells were obtained commercially from Lonza Walkersville, Inc. and were derived from a single 5-year-old donor (de-identified). The HAE cells were cultured and differentiated as described [30]. The ΔNS2/Δ1313/I1314L vaccine candidate (hereafter called ΔNS2) derived from recombinant RSV A2 (Genbank accession number KT992094) was recovered from transfected plasmids as previously described [23]. The four RSV "line 19" (Genbank accession number FJ614813) F mutations (I79M, K191R, T357K and N371Y) were introduced into the F gene of ΔNS2 by site-directed mutagenesis using the GeneArt PLUS kit (GeneArt A14604) and the following primers: I79M-f (5′ TAAGGTAAAATTG ATGAAACAAGAATTAGAT 3′), I79M-r (5′ ATCTAATTCTTGTTTCATCAATTTTACCTTA 3′), K191R-f (5′AGTGTTTTAACCAGCCGGGTGTTAGACCTCAAA 3′), K191R-r (5′ TTT GAGGTCTAACACCCGGCTGGTTAAAACACT 3′), T357K and N371Y-f (5′ TTCCCACAA GCTGAAAAGTGTAAAGTTCAATCAAATCGAGTATTTTGTGACACAATGTATAGTTTAACA TTACCA 3′), T357K and N371Y-r (5′TGGTAATGTTAAACTATACATTGTGTCACAAAA TACTCGATTTGATTGAACTTTACACTTTTCAGCTTGTGGGAA 3′) (sites of codon substitutions underlined). Rescue of ΔNS2 containing the four "line 19" F mutations (hereafter called ΔNS2-L19F-4M) was performed via reverse genetics as previously described [23]. The virus was amplified in Vero cells at 32˚C, and viral titer was determined using RSV immunoplaque assay (below). The RSV A2-L19F virus was kindly provided by Dr. Martin Moore [40]. Recombinant wild-type A2 and A2-L19F viruses were amplified in Vero cells to generate an additional virus control. All full-length cDNA constructs and recovered viruses were subjected to whole-genome sequence analysis of uncloned PCR or RT-PCR amplicons, which confirmed the presence of the introduced mutations and the absence of adventitious or compensatory mutations in all cases.

## RSV plaque assay

Virus titers were determined using an RSV immunoplaque assay. Ten-fold serial dilutions of virus stocks were prepared using cold Dulbecco's Modified Eagle's Medium (DMEM, Sigma D6546) supplemented with 1% GlutaMax (Thermo Fisher Scientific 35-050-061) and 1% antibiotic/antimycotic (Thermo Fisher Scientific 15-240-096). The virus dilutions were inoculated onto confluent monolayers of Vero cells in 24-well cell culture plates. The plates were then incubated at 32˚C and 5% $CO_2$ for 1 hour with gentle agitation every 15 minutes, then overlayed with 1 ml per well of 0.75% methylcellulose (Sigma 64632) dissolved in DMEM supplemented with 2% each of heat-inactivated Fetal Bovine Serum (FBS, Thermo Fisher Scientific 10100147), GlutaMax and antibiotic/antimycotic. The plates were placed in a 32˚C, 5% $CO_2$ incubator for 7 days. The methylcellulose overlay was then removed, and the cells were fixed using 1 ml per well of cold methanol at room temperature for 1 hour. The methanol was removed, then the plates were washed with distilled water, and 200 μl of blocking solution [5% non-fat dry milk in Dulbecco's Phosphate Buffered Saline (DPBS, Gibco 14190–144)] was added per well. The plates were agitated gently using an orbital shaker for 1 hour at room temperature. After removal of the blocking solution, 200 μl of mouse monoclonal antibody to the RSV F protein [RSV3216 (B016); Abcam ab24011] diluted 1:2000 in 1% non-fat dry milk in DPBS was added per well and the plates were gently agitated using an orbital shaker at room temperature for 1 hour. The antibody was removed, then the plates were washed twice with distilled water, and 200 μl of Goat Anti-Mouse IgG H&L (HRP) (Abcam ab6789) diluted 1:2000 in 1% non-fat dry milk in DPBS was added per well. The plates were gently agitated using an orbital shaker at room temperature for 1 hour. The antibody was removed, then the plates were washed twice with distilled water and 200 μl of TrueBlue Peroxidase Substrate

(Fisher Scientific 50-674-28) was added per well. The plates were developed at room temperature with gentle agitation using an orbital shaker until distinct blue plaques were observed (5 to 15 minutes). The plates were washed twice with distilled water and dried at room temperature overnight. The plates were scanned in a plate reader, the plaques were counted, and the viral titer reported as plaque-forming units per ml (PFU/ml). RSV titers in animal study specimens were determined on Vero cells in the same way except that the Abcam antibody was replaced by a mix of three RSV F-specific monoclonal antibodies (MAbs) [41].

## Kinetics of multi-cycle viral replication in Vero and human airway epithelial cells

Replicate cultures of Vero cells at 90% confluence in T25 flasks were infected at a multiplicity of infection (MOI) of 0.1 with recombinant wt RSV, ΔNS2, and ΔNS2-L19F-4M. Following a 1-hour adsorption, the medium was replaced, and the cultures were incubated at 32˚C, 5% $CO_2$. On days 1–7 three, and on days 8–10 two, flasks per virus per day were harvested by scraping cells into the overlying medium. The suspension was vortexed to release cell-associated virus, clarified by low-speed centrifugation and snap-frozen on dry ice. Virus titers were determined later by RSV immunoplaque assay as described above. Only one titer was determined for ΔNS2 on day 10.

HAE cultures differentiated in 6.5 mm Transwell inserts (Corning 3470) were infected, as described [42], at an MOI of 5, with wt RSV strain A2, ΔNS2, and ΔNS2-L19F-4M. Duplicate infections were performed for each construct. On days 1, 3, 5, 7 and 9 post-infection, the apical surfaces of the cultures were washed with media, and the titer of virus in the wash was determined on Vero cells. In the first experiment, the infection and culture were performed at 32˚C, 5% $CO_2$; in the second experiment, the HAE were infected and cultured at 32˚C, 5% $CO_2$ for 2 days, then transferred to 37˚C, 5% $CO_2$.

## ELISA using mAbs specific to pre-F and total F

Virus stocks of RSV A2, A2-L19F, ΔNS2 and ΔNS2-L19F-4M with a titer of >7 $\log_{10}$ PFU per ml were diluted 4-fold in Dulbecco's Phosphate Buffered Saline and 100 μl of this virus suspension was added to duplicate wells in a 96-well Maxisorp plate (Nunc 439454). The plates were sealed and incubated at room temperature overnight. The virus was removed from the plates and the plates were washed once with 200 μl per well of PBST (0.05% Tween20 in Phosphate Buffered Saline), followed by the addition of 150 μl per well of blocking solution (5% bovine serum albumin (BSA, Sigma A3059) in DPBS) and incubation at room temperature for 1 hour. The plates were washed once with 200 μl per well of PBST, followed by the addition of 100 μl per well of serially diluted pre-F-specific antibody D25 [43, 44], specific for antigenic site Ø which is exclusive to the pre-F conformation or Palivizumab (based on monoclonal antibody 1129 [45] which recognizes antigenic site II, shared by pre-F and post-F conformations). D25 and Palivizumab were diluted with 1% BSA in DPBS to 1 μg/ml and serially diluted 4-fold up to a 1024-fold dilution. The plates were incubated at room temperature for 2 hours. The antibodies were removed, and the plates washed with PBST three times. Then 100 μl of europium (Eu)-N1-anti human IgG (Delfia 1244–330) diluted 1:2000 in Delfia assay buffer (Delfia 4002–0010) was added to the wells and incubated at room temperature for 1 hour. The plates were washed with PBST three times, followed by the addition of 100 μl per well of enhancement solution (Delfia 4001–0010). The plates were incubated on a rocker at room temperature for 20 minutes. The time-resolved fluorescence (ex. 340 nm/em. 615 nm) was measured using an Eu-specific program in the Envision 2104 multilabel plate reader. A point in the linear

range was chosen for each virion ELISA with D25 and Palivizumab, and the D25/Palivizumab ratio was calculated.

## Thermal stability

Virus stocks were quickly thawed in a 37°C water bath until a sliver of ice was left. Several vials of each virus were pooled and diluted 1:10 with OptiPRO SFM (Thermo Fisher Scientific 12309019). Aliquots were prepared and incubated at each temperature (4°C, 25°C, 32°C and 37°C). At different days post-incubation, the virus aliquots were vortexed briefly to mix, flash frozen on dry ice and stored at -80°C. At the conclusion of the experiment, the viral titers were determined for all samples as described above.

## Genetic stability in vero cells

Vero cells were infected with ΔNS2-L19F-4M at an MOI of 0.1, and nine further serial passages were performed at 32°C, 5% $CO_2$. At each passage, cells were harvested on day 3 or 4 by scraping into the cell culture medium. The suspension was vortexed to release cell-associated virus, clarified by low-speed centrifugation, and used to infect fresh Vero cells at an estimated MOI of 0.1. The remaining supernatant was flash-frozen.

The whole genome sequence of the passage 10 material was then determined. In brief, viral RNA was extracted using the QIAamp Viral RNA kit (Qiagen 52904) and cDNA generated using the Protoscript First strand cDNA synthesis kit (New England Biolabs E6300). The genome was amplified in 5 reactions using the PrimeSTAR GXL DNA polymerase (Takara/Clontech #R050A), and the gel purified DNA products sent to Genewiz for sequencing. The contigs were assembled and the consensus sequence analyzed for mutations using Vector NTI Software (Thermo Fisher Scientific).

## Attenuation and immunogenicity in African green monkeys (AGMs)

Animal studies were carried out in accordance with the Guide for Care and Use of Laboratory Animals of the National Institutes of Health and were approved by the NIAID Animal Care and Use Committee. The animals were housed in accord with NIAID DIR Animal Care and Use Program Policy on Social Housing of NHPs. The animals were on a special enrichment program (additional objects for manipulation, perches, etc.). Juvenile to young adult AGMs (*Chlorocebus aethiops*) were inoculated with the indicated viruses (four animals per group). Two groups were inoculated by the combined intranasal and intratracheal (IN/IT) route at a dose per site of $10^6$ PFU in 1 ml of L-15 medium (Thermo Fisher Scientific 11415064) (total dose per animal: 2 x $10^6$ PFU). Following inoculation, nasopharyngeal swabs (NP) were collected daily from days 1 through 10 and on days 12 and 14. Tracheal lavages (TL) were collected every other day for 14 days. Samples were flash-frozen on dry ice and stored at -80°C. Procedures were performed under ketamine anesthesia. In this survival study no animals were sacrificed. Upon study termination the animals were returned to the colony for use in further research. Virus titers in NP and TL aspirates were determined by RSV immunoplaque assay on Vero cells at 32°C as described above. Sera were collected prior to inoculation and on days 21 and 28. Results were compared to historic controls from AGMs from the same origin and age group that had been inoculated with recombinant wt RSV A2 (Genbank KT992094) by the IN/IT route, following the same protocol and SOPs. The only difference was that NP and TL were not collected on day 12 in the previous study. Statistical analyses were performed by 2-way ANOVA with Tukey's post-hoc analysis (GraphPad Prism version 9, GraphPad Software, San Diego, California USA).

Serum RSV-neutralizing antibody titers were measured by 60% plaque reduction neutralization (PRN) assays on Vero cells [46] using RSV expressing green fluorescent protein (GFP). The assays were performed in the presence of 5% added guinea pig complement (Cedarlane, Burlington, NC). Plaques were visualized by GFP expression using a Typhoon imager (GE Healthcare, Piscataway, NJ).

## Results

### Generation of the ΔNS2-L19F-4M vaccine candidate by reverse genetics

Based on its safety and immunogenicity in a previous phase 1 study, the RSV ΔNS2 LAV candidate has been evaluated in a recently completed phase 1/2 clinical trial in infants and toddlers. To improve the ease of manufacture of ΔNS2 and enhance its immunogenicity, we used reverse genetics to transfer four amino acid changes into the F protein of ΔNS2 to generate the modified candidate ΔNS2-L19F-4M (Fig 1). These substitutions were derived from the F protein of RSV strain "line 19" (I79M, K191R, T357K, N371Y). In previous studies, these amino acid assignments were shown to enhance pre-F levels and thermal stability of RSV. The ΔNS2-L19F-4M construct was rescued from cDNA by reverse genetics, and ΔNS2 and ΔNS2-L19F-4M were amplified in Vero cells at 32°C, the permissive temperature for these temperature-sensitive viruses. Working pools were generated. Whole genome sequencing confirmed that the sequences of the viral genomes were identical to the cDNA each was derived from.

### The ΔNS2-L19F-4M virus replicated with the same efficiency as the parental ΔNS2 virus in Vero cells

To evaluate the growth of the modified vaccine candidate ΔNS2-L19F-4M in cell culture over multiple replication cycles, Vero cells were infected at an MOI of 0.1, and incubated at 32°C, 5% $CO_2$. Virus was harvested daily until day 10 and after the conclusion of the experiment, viral titers were determined using an RSV immunoplaque assay. ΔNS2 reached titers above $10^7$ PFU/mL by day 3, with slightly higher peak titers by day 5 or 6 (Fig 2). We found that ΔNS2-L19F-4M growth was delayed, reaching titers above $10^7$ PFU/mL by day 7, but its peak titer was similar to that of the parental virus.

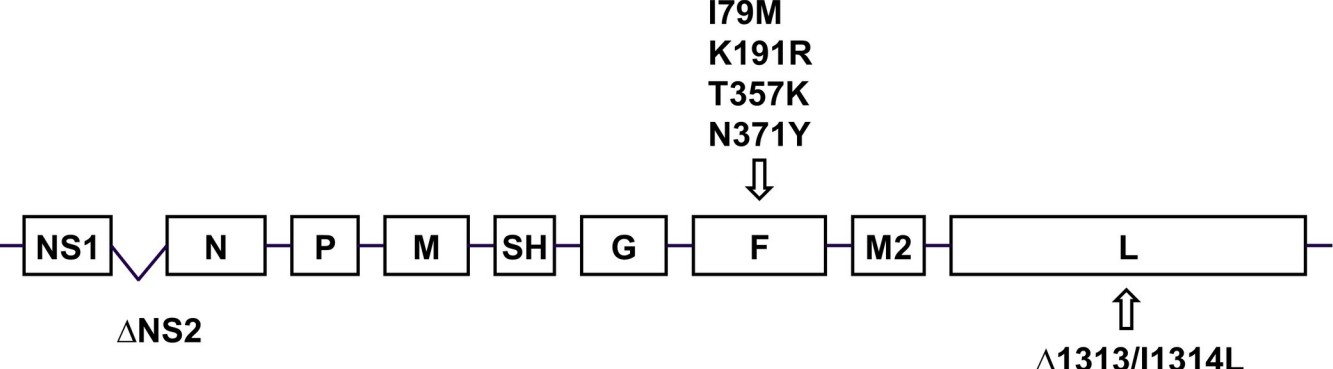

**Fig 1. Genome map of ΔNS2-L19F-4M vaccine candidate.** Genes encoding separate mRNAs are indicated by boxes shown in the 3´-to-5´ order. The positions of the deletion of the NS2 gene, the deletion of codon 1313 in the L gene, and the missense mutation at codon 1314 in the L gene are shown. Also shown are the positions of the "line 19" F mutations I79M, K191R, T357K and N371Y (named by F protein amino acid position preceded by the A2 assignment and followed by the "line 19" assignment).

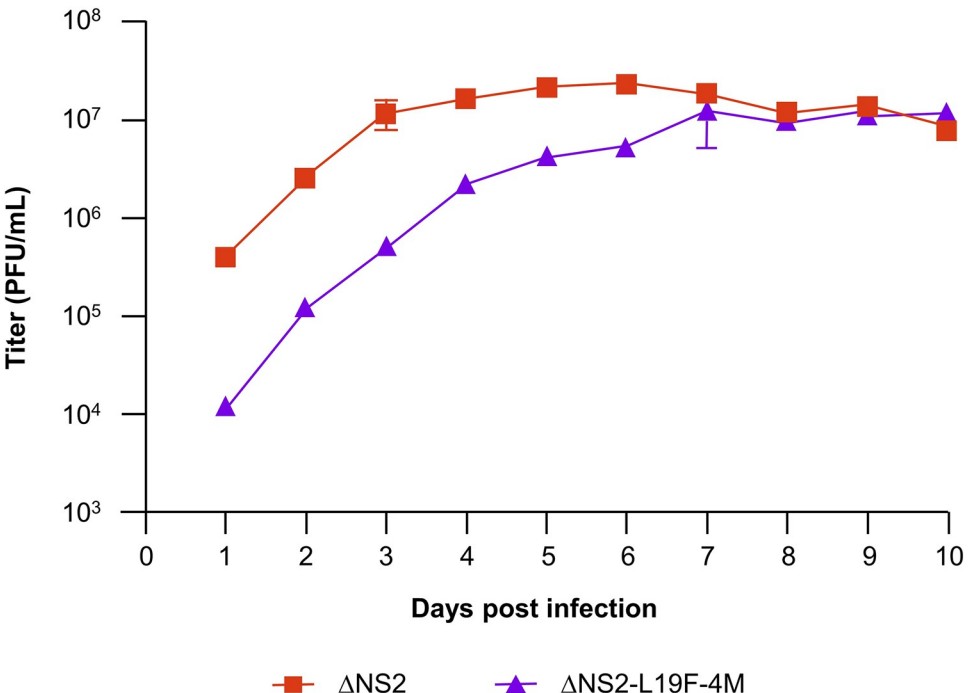

**Fig 2. Multicycle growth of the ΔNS2-L19F-4M and ΔNS2 RSVs in Vero cells.** Vero cells were infected at MOI of 0.1 with either ΔNS2-L19F-4M or the parental ΔNS2 virus and incubated at 32°C, 5% $CO_2$. For each virus, 2–3 flasks were harvested daily and frozen for analysis later. Viral titers were determined using an RSV immunoplaque assay. Error bars represent standard deviation from triplicate flasks on days 1–7 and duplicate flasks on days 8–10.

## ΔNS2-L19F-4M virions exhibited a higher content of pre-F protein than parental ΔNS2 virions

To determine whether ΔNS2-L19F-4M has an increased amount of pre-F protein compared to the ΔNS2 parent, we compared the ratios of pre-F to total F in virus stocks of RSV A2, A2-L19F, ΔNS2-L19F-4M and ΔNS2 using ELISAs to measure conformation-specific antibody binding. The levels of pre-F protein present in these virus suspensions were measured using D25, a human MAb that recognizes an antigenic site unique to pre-F (site Ø). The total F protein levels were evaluated by using Palivizumab, a humanized MAb that recognizes an antigenic site common to the pre-F and post-F conformations (site II). We interpreted the ratio of binding of D25 versus Palivizumab as a measure of the ratio of pre-F protein to total F protein, although it is recognized that this is a ratio of ELISA signals rather than an actual determination of protein mass.

Comparing the non-attenuated viruses, A2-L19F had a ratio of pre-F/total F that was 1.2- to 1.4-fold higher than for RSV A2 (Fig 3A). Unexpectedly, the pre-F/total F ratio for ΔNS2-L19F-4M was 4–5.5-fold higher than for ΔNS2 (Fig 3B). These findings showed that inclusion of the four "line 19" F mutations into wt RSV A2 provided a modest increase in the proportion of pre-F protein, whereas inclusion of these mutations into the attenuated ΔNS2 virus unexpectedly provided a much larger increase.

## The ΔNS2-L19F-4M virus has enhanced thermal stability

RSV infectivity is readily lost during handling and storage, especially at warmer temperatures. A major factor is thought to be the premature and irreversible conversion of pre-F to post-F

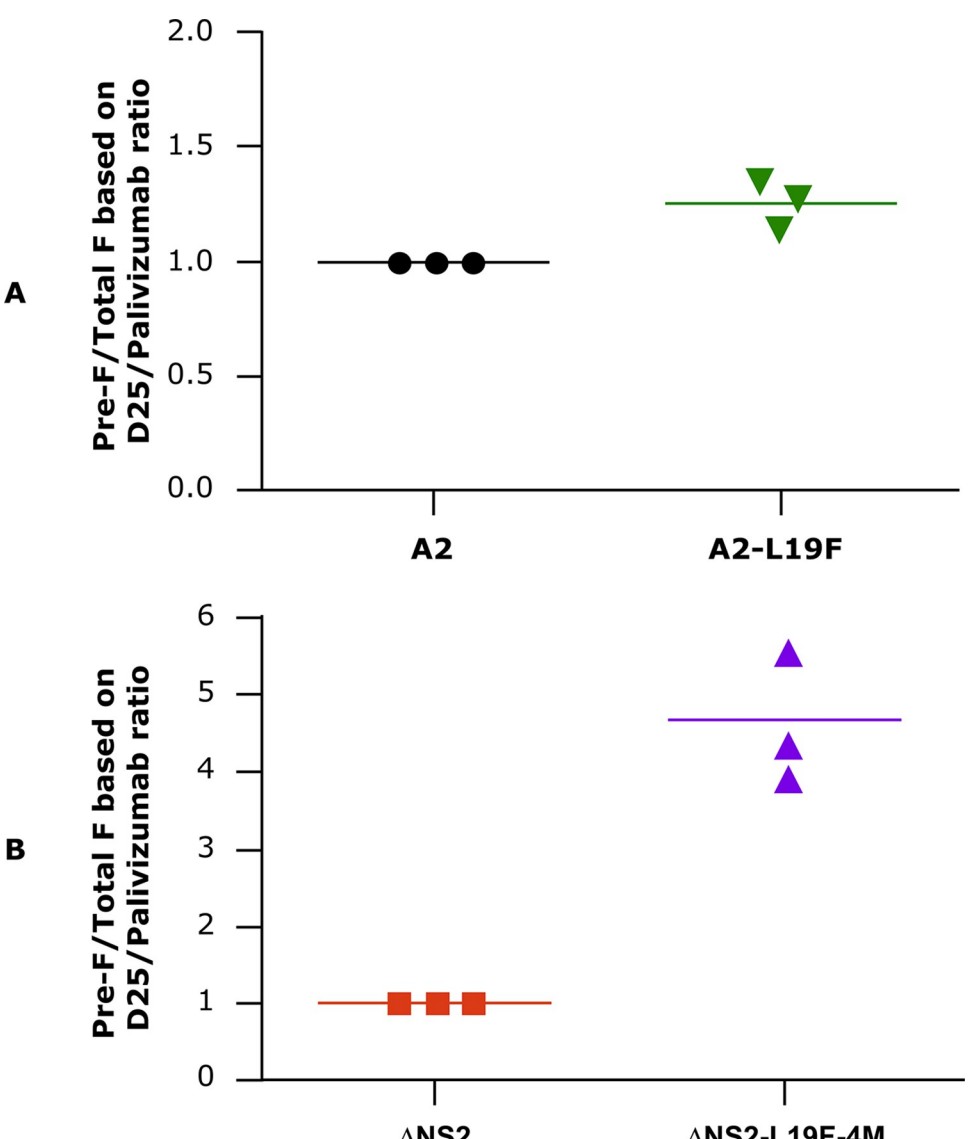

**Fig 3. Pre-F/total F ratio on RSV virions.** The pre-F levels in recombinant virus were determined by ELISA. Plates were coated with virus stocks of RSV A2, A2-L19F, ΔNS2 and ΔNS2-L19F-4M. Plates were incubated with D25 MAb to quantify pre-F and with Palivizumab to quantify total F protein content. (A, B) Ratios of the ELISA signal for D25 compared to that of Palivizumab. (A) Ratios for RSV A2 and A2-L19F, normalized to RSV A2 as 1.0. (B) Ratios for ΔNS2 and ΔNS2-L19F-4M, normalized to ΔNS2 as 1.0. The results of 3 independent experiments are shown.

[47], resulting in loss of infectivity and vaccine potency. As ΔNS2-L19F-4M exhibited a higher pre-F/total F ratio than ΔNS2, we next investigated if this translates to enhanced thermal stability and stability of the virus titers over time. Virus stocks were incubated at 4˚C, 25˚C, and 32˚C over a period of 28 days, and at 37˚C over a period of 7 days, and viral titers were determined using an RSV immunoplaque assay.

At 4˚C, both ΔNS2 viruses maintained their stability over long periods of time; virus titers slowly declined by about 10- to 50-fold over 28 days (Fig 4). For ΔNS2-L19F-4M, the decline in titers at 4˚C was less than 10-fold, and kinetics of the decline were slightly slower compared to ΔNS2. At higher temperatures (25˚C, 32˚C and 37˚C), the stability of both viruses declined more drastically and rapidly than at 4˚C, with ΔNS2 losing infectivity more rapidly than

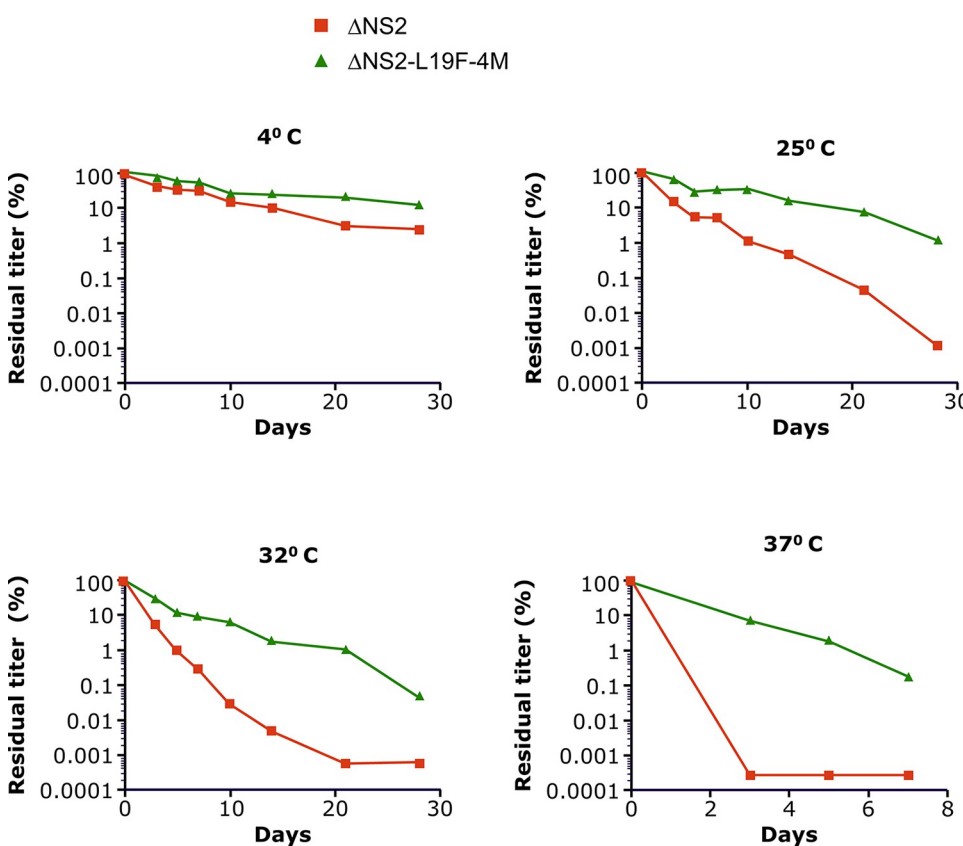

**Fig 4. Thermal stability of ΔNS2-L19F-4M and ΔNS2.** Virus stocks of ΔNS2-L19F-4M or the ΔNS2 parent were incubated at the indicated temperature (4°C, 25°C, 32°C and 37°C). At the indicated days, aliquots were frozen on dry ice and stored at -80°C, and virus titers were determined later by immunoplaque assay. Viral titers were expressed as a percentage of the initial titer.

ΔNS2-L19F-4M (Fig 4). For example, at 32°C, titers of ΔNS2 and ΔNS2-L19F-4M declined over 28 days by factors of about $10^6$ and $10^4$, respectively, a 100-fold difference between the two viruses. ΔNS2-L19F-4M demonstrated higher stability at 4°C or notably superior stability at 25°C, 32°C and 37°C than that of ΔNS2, indicating that ΔNS2-L19F-4M is more thermostable than ΔNS2 at every temperature tested. Thus, the L19F-4M substitutions substantially improved the stability of vaccine infectivity over time.

## The four "line 19" F mutations were genetically stable in the ΔNS2-L19F-4M virus

To ensure that the four "line 19" F mutations will be retained during clinical lot production, the genetic stability of the ΔNS2-L19F-4M F gene was assessed. The ΔNS2-L19F-4M virus was subjected to ten rounds of serial passage in Vero cells at 32°C using an estimated MOI of 0.1. The passaging was via clarified cell lysate, and viral RNA was extracted from the passage 10 lysate for whole genome sequencing. None of the four "line 19" F mutations showed any sign of reversion or instability. A partial shift from lysine to asparagine at amino acid position 75 of the F gene was observed with the chromatograph peak of the mutant nucleotide equivalent or slightly higher than the wild type in all 6 reads through the position. No other mutations in the F protein were detected. When the plaque morphology of the passage 10 material was compared to that of the original stock, the passage 10 plaques were overall slightly larger (S1 Fig).

## ΔNS2-L19F-4M showed comparable attenuation as the parental ΔNS2 virus in cultures of primary human airway epithelium (HAE)

To determine if the addition of the four "line 19" F mutations affected the attenuation profile of ΔNS2-L19F-4M, we initially utilized the primary differentiated human airway epithelial (HAE) cell system. This *ex vivo* model recapitulates many features of the *in vivo* human airway including a stratified cell layer, polarization, actively beating cilia, mucus production, and the binding of RSV to the CX3CR1 receptor to initiate entry [28–30]. Unlike monolayer cultures of immortalized cells, HAE cultures can provide a facsimile of *in vivo* attenuation and have been utilized to characterize the attenuation of vaccine candidates [42, 48, 49].

Fully differentiated HAE cultures, prepared from cells from a 5-year-old donor, were infected in duplicate with wild type RSV strain A2 or the vaccine candidates ΔNS2 and ΔNS2-L19F-4M. On days 1, 3, 5, 7 and 9 post-infection, the apical surfaces of the cultures were washed with media and the titer of virus in the wash was determined on Vero cells. Two experiments were performed. In the first experiment, the infection and culture were performed at 32˚C (Fig 5A). In the second experiment, the HAE were infected and cultured at 32˚C for 2 days, then transferred to 37˚C to better simulate the conditions encountered *in vivo* as RSV spreads from the upper to the lower airways (Fig 5B). In both experiments, the attenuation profile of ΔNS2-L19F-4M was similar to that of the parental ΔNS2, with the peak titers of both candidates at least 100-fold lower than the wild type A2. In the second experiment, the titers of both candidates were further attenuated compared to growth at 32˚C reflecting the temperature sensitivity induced by the Δ1313 mutation [23].

## ΔNS2-L19F-4M and the parental ΔNS2 were similarly attenuated and immunogenic in African green monkeys (AGMs)

The replication of ΔNS2 and ΔNS2-L19F-4M was evaluated in AGMs. For comparison, historic controls were included from a previous study performed in AGMs of the same age group and background, inoculated with recombinant wild type (wt) RSV A2 (same route and dose), following the same procedures as in the present study, except that D12 NP and TL samples were not collected. Even though AGMs are only semi-permissive for RSV, they are susceptible to RSV and represent the most relevant preclinical animal model available. The body temperature of AGMs is about 1˚C higher than that of humans. Following infection by the intranasal/intratracheal route, robust shedding of RSV A2 was detectable in NP specimens and TL. Specifically, in NP specimens (Fig 6A), peak titers of 3.9 to 5.2 $\log_{10}$ PFU per ml were detected in individual animals between days 4 and 7 post-immunization (geometric mean peak titers (GMT) independent of study day: 4.2 $\log_{10}$ per ml). In TL specimens (Fig 6B), peak titers of 3.0 to 4.7 $\log_{10}$ PFU per ml were detected between days 6 and 8 (GMT independent of study day: 4.1 $\log_{10}$ per ml). Replication of both vaccine candidates ΔNS2 and ΔNS2-L19F-4M was also detectable over several days in NP specimens and TL. In the upper airways, peak titers of shed vaccine ranged between 2.9 to 4.7 $\log_{10}$ (ΔNS2) and 3.6 to 4.8 $\log_{10}$ (ΔNS2-L19F-4M). The peaks of viral shedding of both vaccine candidates were on days 9 and 10, substantially later than those in the RSV A2 group. However, GMTs, independent of study day, of ΔNS2 and ΔNS2-L19F-4M were about 2.5- and 5-fold lower, respectively, than those of wt RSV.

In the lower airways, ΔNS2 and ΔNS2-L19F-4M vaccine shedding was sporadic; in groups inoculated with ΔNS2 or ΔNS2-L19F-4M by the IN/IT route, only 3 of 4 animals had virus detectable in the TL. GMTs of ΔNS2 and ΔNS2-L19F-4M, independent of study day, were 1.8 $\log_{10}$ and 1.6 $\log_{10}$ PFU per mL, respectively, about 200- and 300-fold lower than those of RSV A2. While RSV A2 shedding in TL peaked at day 6, that of ΔNS2 and ΔNS2-L19F-4M peaked at day 10. Thus, compared to RSV A2, replication of the ΔNS2 and ΔNS2-L19F-4M candidates

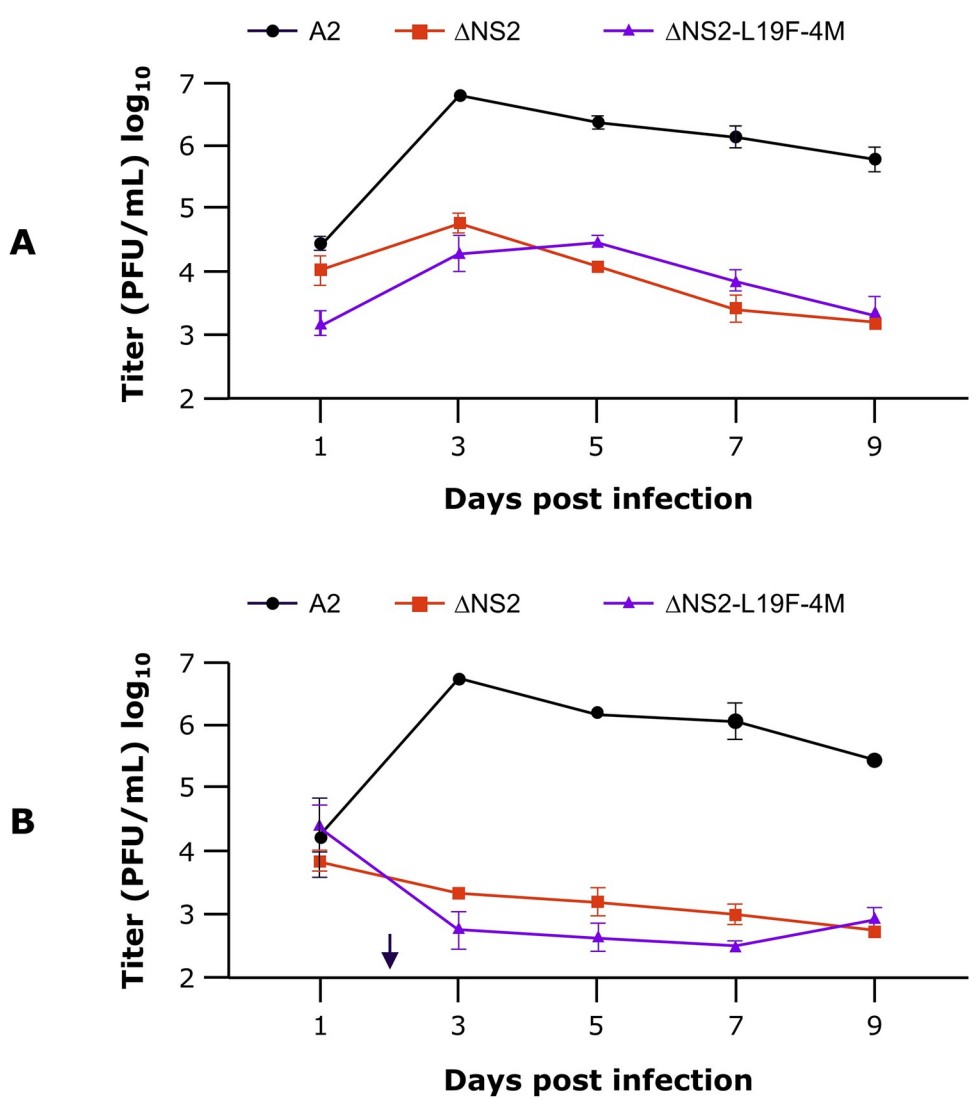

**Fig 5. Replication in human airway epithelial (HAE) cells.** HAE cultures were infected in duplicate at an MOI of 5 and cultured at 32˚C, 5% $CO_2$ for 9 days (A) or transferred to 37˚C, 5% $CO_2$ on day 2 and incubated for 7 more days (B). The arrow in (B) indicates time of transfer from 32˚C to 37˚C. Viral titers were determined from apical wash samples collected on the indicated days, flash-frozen, and quantified later by immunoplaque assay. Error bars represent standard deviation from duplicate infections.

was restricted in AGMs, and the restriction was much more pronounced in the lower airways, reflecting the temperature sensitivity of ΔNS2 and ΔNS2-L19F-4M conferred by the Δ1313/ I1314L mutations in the L gene of these viruses. Shedding of ΔNS2 and ΔNS2-L19F-4M and replication in the AGM of ΔNS2 and ΔNS2-L19F-4M were similar, suggesting that the L19-4M substitutions did not have a detectable effect on *in vivo* replication. The animals were monitored for clinical symptoms over the duration of the study, with no significant changes from baseline detectable, confirming that RSV A2 does not cause any clinical disease in AGMs.

We also compared the immunogenicity of RSV A2 and the vaccine candidates in these animals (Fig 6C). On days 21 and 28, mean RSV serum neutralizing titers of 7.2 and 7.5 $\log_2$ were detected by $PRNT_{60}$ in RSV A2 inoculated animals. In ΔNS2 and ΔNS2-L19F-4M immunized animals, RSV $PRNT_{60}$ titers were similar (7.8 and 8.9 $\log_2$ on days 21 and 28 in the ΔNS2

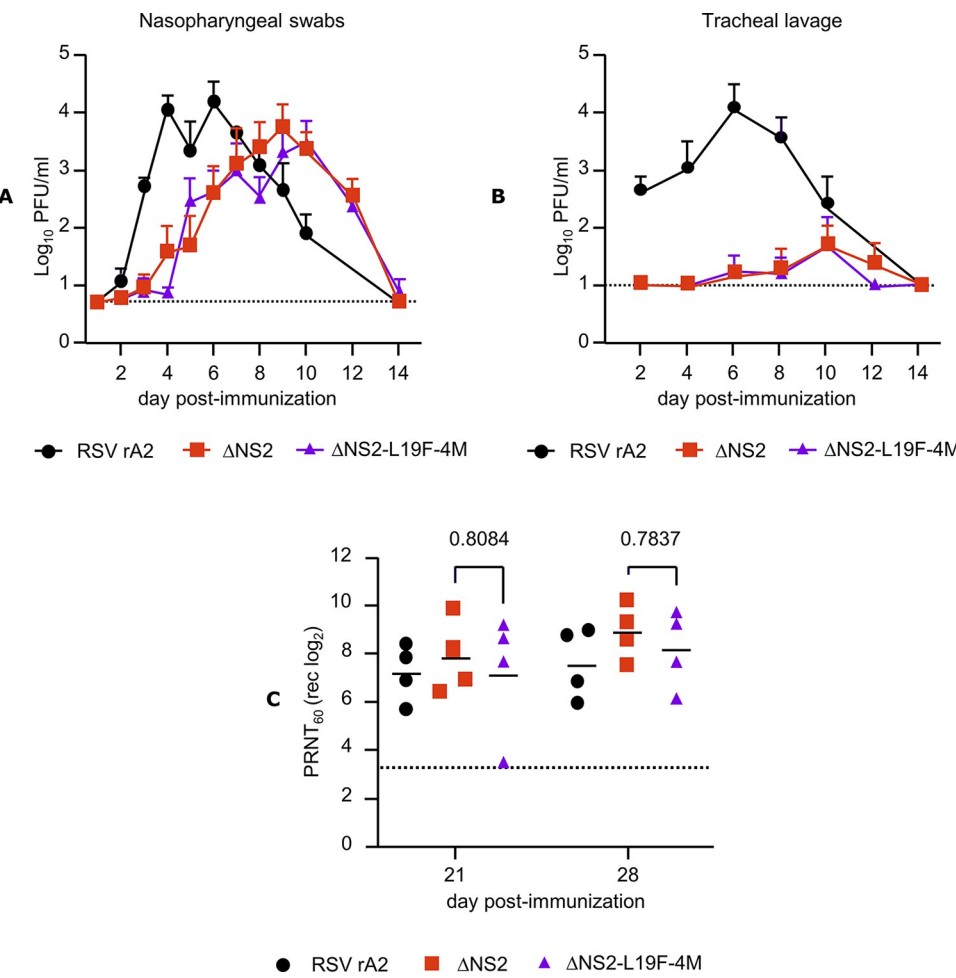

**Fig 6. Attenuation and immunogenicity in African green monkeys (AGMs).** Monkeys were inoculated via the combined intranasal (IN) and intratracheal (IT) route with $10^6$ PFU of each virus in a 1 ml inoculum per site (total dose = $2 \times 10^6$ PFU per AGM). Nasopharyngeal swabs were taken on days 1 through 10 and on days 12 and 14, and tracheal lavages were performed every other day through day 14. Virus shedding in nasopharyngeal swabs (A) and tracheal lavage (B) were measured using an RSV immunoplaque assay on Vero cells. The lower limits of virus detection were 0.7 $\log_{10}$ (A) and 1.0 $\log_{10}$ PFU per mL (B), respectively (dashed lines). (C) Sera were collected on days 21 and 28 post-immunization, and RSV-neutralizing titers, plotted as $1/\log_2$ were determined using a 60% plaque reduction neutralization test ($PRNT_{60}$) in presence of complement. The lower limit of detection was 3.3 $\log_2$ (dashed line) (C). Statistical analyses were performed by 2-way ANOVA with Tukey's post-hoc analysis (GraphPad Prism, version 9, GraphPad software, San Diego, California, USA).

group, and 7.1 and 8.1 $\log_2$ in the ΔNS2-L19F-4M group on these days). Thus, despite the lower level of replication of ΔNS2 and ΔNS2-L19F-4M, the vaccine candidates induced RSV-neutralizing serum antibody titers comparable to RSV A2 in AGMs.

## Discussion

We have successfully incorporated four "line 19" F mutations into our lead live-attenuated vaccine ΔNS2 to generate the modified version ΔNS2-L19F-4M. Consistent with the results of Stobart et al. [40], the ΔNS2-L19F-4M virus exhibited an enhanced ratio of pre-F/total F as compared to ΔNS2, suggesting an increased stability of the pre-F conformation on the virus surface. These data support the idea that the four "line 19" F mutations have the capability to transiently stabilize the pre-F form of the RSV F protein. Interestingly, this effect was

substantially greater in the ΔNS2 backbone compared to the wild type backbone. The reason for greater stabilization in the ΔNS2 backbone compared to wt is unknown. An increase in the content of pre-F protein in a vaccine candidate could be desirable because of increased infectivity and immunogenicity, but also raises the possibility of effects on attenuation *in vivo*.

The infectivity of the ΔNS2-L19F-4M virus was more thermostable than the ΔNS2 parent at all temperatures tested, and the difference was more pronounced at higher temperatures (i.e., 37˚C). Thermal stability at 4˚C is important for retention of vaccine potency at a convenient cold chain temperature, whereas stability at 25˚C, 32˚C and 37˚C may allow for vaccine transport without refrigeration. In addition, stability at 32˚C and 37˚C is relevant for the propagation and harvesting of virus stocks, and, importantly, these temperatures reflect physiological conditions in the upper (32˚C) and lower (37˚C) airways. This could have important implications for a live RSV vaccine, as the virus would be more resistant to losses during manufacture, packaging, storage, transportation, distribution, and use. RSV typically requires snap-freezing, storage at -80˚C and careful thawing. It would be highly desirable to be able to handle and store the vaccine at more accessible temperatures. In particular, this could greatly facilitate worldwide distribution of the vaccine, including locations where cold storage is more limited. We presume that the increased thermostability of ΔNS2-L19F-4M infectivity is due to increased stability of the pre-F conformation, although this was not investigated.

The kinetics and efficiency of replication of the ΔNS2-L19F-4M virus in Vero cells, which is the substrate for vaccine manufacture, was very similar to that of the ΔNS2 virus. This also mirrors the results of Stobart et al. [40]. In addition, the four "line 19" F mutations were genetically stable over 10 passages in Vero cells. However, an adventitious mutation K75N in the F protein was detected at passage 10, and the plaques from passage 10 material were slightly larger. It would be interesting to conduct a detailed study to determine if this mutation has any further phenotypic effects and negative impact on the stability of the four "line 19" F mutations *in vivo*.

It was established previously that RSV LAV candidates exhibit growth in HAE that corresponds to their level of replication in children [49]. For RSV LAV candidates in general, the level of replication in the target host (or HAE cultures) is a principal predictor of attenuation and tolerability. In this study, we also conducted HAE replication experiments and demonstrated that the four "line 19" F mutations could be incorporated in the RSV ΔNS2 live-attenuated vaccine candidate without affecting its replication profile in HAE, predicting that the attenuation phenotype was not altered by the inclusion of the L19-4M substitutions.

The attenuation and immunogenicity of ΔNS2-L19F-4M compared to ΔNS2 and wt RSV was further evaluated in AGMs. Consistent with the findings for HAE cultures, the level of replication of ΔNS2-L19F-4M in AGMs was similar to that of the parental ΔNS2 virus. This suggests that the attenuation of the ΔNS2 vaccine candidate was unaffected by the increased content of pre-F protein. It would have been anticipated that the increased content of pre-F protein in ΔNS2-L19F-4M would induce an increased level of serum RSV-neutralizing antibodies, since the F protein is the most important RSV neutralization antigen and the pre-F conformation is the most immunogenic form of F. However, this was not observed: the titers of serum RSV-neutralizing antibodies induced in AGMs by ΔNS2-L19F-4M were indistinguishable from those induced by its ΔNS2 parent.

It may be that the AGM model would not have been sufficiently sensitive to reliably detect a modest difference in immunogenicity between ΔNS2-L19F-4M and ΔNS2. This is suggested by the observation that the antibody titers of both ΔNS2-L19F-4M and ΔNS2 in AGMs were indistinguishable from those of a historic control of wt recombinant RSV A2 even though the wt virus replicated to a higher titer, especially in the tracheal lavages. On the other hand, differences in immunogenicity associated with differences in pre-F content had been observed in

previous studies in which live-attenuated bovine/human parainfluenza virus (B/HPIV3) vectors were used to compare the immunogenicity of the wt RSV F protein to several versions that had been stabilized in the pre-F conformation by the DS-Cav substitutions [34, 36, 50, 51]. These vector-expressed, DS-Cav-stabilized versions of the RSV F protein elicited higher titers of RSV-neutralizing serum antibodies than wt F protein in hamsters and rhesus monkeys, and also induced increased titers of high-quality antibodies with the ability to neutralize RSV in absence of complement. However, in these previous studies the F protein was strongly stabilized in the pre-F conformation such that these F proteins were not functional in fusion, whereas in the present study the "line 19" mutations would provide only transient stabilization that permitted fusion to occur.

This study has shown that introduction of four amino acid changes into the F protein of ΔNS2 virus led to increased stability of infectious virus without changing the attenuation profile or antibody response. The improved stability resulting from the introduction of the line 19 F mutations could offer significant advantages for the manufacturing, storage, transportation, and utilization of live attenuated RSV vaccines, ultimately leading to cost reduction and increased vaccine accessibility. Altogether, the results of this study support the inclusion of these substitutions into live-attenuated vaccine candidates for future evaluation in clinical trials.

## Supporting information

**S1 Fig. Plaque morphology of ΔNS2-L19F-4M and ΔNS2 before and after 10 rounds of serial passage.** Images of representative wells from titrations of (A) original ΔNS2-L19F-4M material and (B) harvest from the 10<sup>th</sup> round of serial passage. Plaques were grown under overlay for 7 days at 32˚C.
(TIF)

**S1 File. The values behind the means and standard deviations used to build graphs are provided as tables in the supplementary information file.** S1 Table. Data used to build Fig 2. S2 Table. Data used to build Fig 3A. S3 Table. Data used to build Fig 3B. S4 Table. Data used to build Fig 5A. S5 Table. Data used to build Fig 5B. S6 Table. Data used to build Fig 6A. S7 Table. Data used to build Fig 6B. S8 Table. Data used to build Fig 6C.
(PDF)

## Acknowledgments

We gratefully acknowledge Dr. Rachel Groppo for technical assistance. We thank Dr. Martin Moore for providing the RSV A2-L19F virus.

## Author Contributions

**Conceptualization:** Judith Alamares-Sapuay, Peter L. Collins, Ursula J. Buchholz, Linong Zhang.

**Data curation:** Judith Alamares-Sapuay, Michael Kishko, Charles Lai, Richard Herbert, Ashley Castens, Joanna Swerczek, Cindy Luongo, Lijuan Yang, Peter L. Collins, Ursula J. Buchholz, Linong Zhang.

**Formal analysis:** Judith Alamares-Sapuay, Michael Kishko, Peter L. Collins, Ursula J. Buchholz, Linong Zhang.

**Investigation:** Judith Alamares-Sapuay, Michael Kishko, Charles Lai, Richard Herbert, Ashley Castens, Joanna Swerczek, Cindy Luongo, Lijuan Yang, Peter L. Collins, Ursula J. Buchholz, Linong Zhang.

**Methodology:** Judith Alamares-Sapuay, Michael Kishko, Charles Lai, Richard Herbert, Ashley Castens, Joanna Swerczek, Cindy Luongo, Lijuan Yang, Ursula J. Buchholz, Linong Zhang.

**Project administration:** Linong Zhang.

**Resources:** Linong Zhang.

**Software:** Linong Zhang.

**Supervision:** Mark Parrington, Simon Delagrave, Peter L. Collins, Ursula J. Buchholz, Linong Zhang.

**Validation:** Peter L. Collins, Ursula J. Buchholz, Linong Zhang.

**Visualization:** Judith Alamares-Sapuay, Peter L. Collins, Ursula J. Buchholz, Linong Zhang.

**Writing – original draft:** Judith Alamares-Sapuay, Peter L. Collins, Ursula J. Buchholz, Linong Zhang.

**Writing – review & editing:** Michael Kishko, Mark Parrington, Simon Delagrave, Peter L. Collins, Ursula J. Buchholz, Linong Zhang.

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
