## [Editor Report · Decision Letter 0]

30 Jan 2024

PONE-D-24-02628Mutations in the F protein of the live-attenuated respiratory syncytial virus vaccine candidate ∆NS2/∆1313/I1314L increase the stability of infectivity and content of prefusion F proteinPLOS ONE

Dear Dr. Zhang,

Thank you for submitting your manuscript to PLOS ONE. After careful consideration, we feel that it has merit but does not fully meet PLOS ONE’s publication criteria as it currently stands. Therefore, we invite you to submit a revised version of the manuscript that addresses the points raised during the review process.

The manuscript was clear and concise, and the studies were done well. There were a few minor questions to consider:

Line 192: please clarify why the cultures were transferred from 32C to 37C after 2 days – did that alter cell numbers appreciably?Line 275: please indicate/comment/reference why ΔNS2-L19F-4M growth was delayedLine 362: please comment on why both vaccine candidates' viral shedding peaks were substantially later (d 9,10) than those in the RSV A2 group. Does cell cytotoxicity correlate with this difference?

Please speculate how the adventitious mutation K75N in the F protein affects plaque size/spread

We look forward to receiving your revised manuscript.

Kind regards,

Ralph A. Tripp

Academic Editor

PLOS ONE

Journal Requirements:

2. In order to comply with PLOS ONE's guidelines for non-human primate experiments (http://journals.plos.org/plosone/s/submission-guidelines#loc-non-human-primates), please provide additional details regarding housing conditions, feeding regimens, environmental enrichment, and all relevant steps taken to alleviate suffering. Also indicate how often animal care staff monitored the health and well-being of the animals and the criteria used to make such assessments. Lastly, specify the disposition of animals at the end of the study (e.g. euthanasia, returned to home colony, etc.).

3. In your Methods section, please include a comment about the state of the animals following this research. Were they euthanized or housed for use in further research? If any animals were sacrificed by the authors, please include the method of euthanasia and describe any efforts that were undertaken to reduce animal suffering.

Additional Editor comments:

The manuscript was clear and concise, and the studies were done well. There were a few minor questions to consider:

• Line 192: please clarify why the cultures were transferred from 32C to 37C after 2 days – did that alter cell numbers appreciably?

• Line 275: please indicate/comment/reference why ΔNS2-L19F-4M growth was delayed

• Line 362: please comment on why both vaccine candidates' viral shedding peaks were substantially later (d 9,10) than those in the RSV A2 group. Does cell cytotoxicity correlate with this difference?

• Please speculate how the adventitious mutation K75N in the F protein affects plaque size/spread

---

## [Author Response · Author response to Decision Letter 0]

16 Feb 2024

Dear Editors,

First, I would like to express our sincere gratitude to the reviewers for their invaluable time in evaluating our manuscript: PONE-D-24-02628 “Mutations in the F protein of the live-attenuated respiratory syncytial virus vaccine candidate ∆NS2/∆1313/I1314L increase the stability of infectivity and content of prefusion F protein”. 

Regarding all comments received, we have thoroughly addressed each point and provided comprehensive responses, which can be found below. Furthermore, we have revised the manuscript to incorporate the additional requirements outlined in the decision letter from PLOS ONE. We are confident that the revised manuscript now meets all necessary criteria for publication in PLOS ONE. 

Sincerely,

Dr. Linong Zhang

Senior expert scientist, Sanofi

Line 192: please clarify why the cultures were transferred from 32C to 37C after 2 days – did that alter cell numbers appreciably?

• Answer: 

o As explained in lines 385 – 387 “In the second experiment, the HAE were infected and cultured at 32°C for 2 days, then transferred to 37°C to better simulate the conditions encountered in vivo as RSV spreads from the upper to the lower airways”.

o The HAE cultures were fully differentiated before infection (line 380). Fully differentiated HAE cultures are 100% confluent and there is no further cell division during the infection time course. Therefore, transfer of the cultures from 32°C to 37°C does not alter cell numbers.

• Line 275: please indicate/comment/reference why ΔNS2-L19F-4M growth was delayed

• Answer: 

o The delay in viral growth could be due to the slower spread of infection by ΔNS2-L19F-4M, possibly influenced by a more stable form of the fusion protein responsible for mediating virus infection in Vero cells. However, despite this delay, ΔNS2-L19F-4M ultimately reached the same level of virus peak titers as the parental virus. 

• Line 362: please comment on why both vaccine candidates' viral shedding peaks were substantially later (d 9,10) than those in the RSV A2 group. Does cell cytotoxicity correlate with this difference?

• Answer: 

o The delay in peak shedding for both vaccine candidates is primarily attributed to the lack of IFN antagonist effect mediated by the NS2 gene, and to the deletion of codon 1313 of the RSV polymerase protein L. The NS2 gene was intentionally deleted from the two attenuated strains. RSV with deletion of the NS2 gene grows more slowly and to lower titers in interferon competent cells, e.g., A549 or Hep-2 (N Teng, P L Collins. Altered growth characteristics of recombinant respiratory syncytial viruses which do not produce NS2 protein. J Virol. 1999 Jan;73(1):466-73). Both vaccine candidates are also attenuated by a deletion of codon 1313 of the polymerase L ORF, which is attenuating and confers temperature sensitivity to RSV. When grown at the fully permissive temperature of 32°C in Vero cells, which do not have the ability to produce Type I interferon, the replication kinetics of RSV/�NS2/�1313/I1314L are highly similar to those of RSV A2 (which is beneficial for vaccine manufacture). However, in animal models and clinical studies (C K Cunningham, R A Karron Evaluation of Recombinant Live-Attenuated Respiratory Syncytial Virus (RSV) Vaccines RSV/ΔNS2/Δ1313/I1314L and RSV/276 in RSV-Seronegative Children. J Infect Dis. 2022 Dec 13;226(12):2069-2078), shedding of RSV/�NS2/�1313/I1314L peaks later than wildtype RSV A2 would. This is one of the key features of attenuation.

• Please speculate how the adventitious mutation K75N in the F protein affects plaque size/spread

• Answer: 

o We speculate that the K75N mutation may favor more efficient fusion activity, thereby triggering fusion events leading to faster viral spread and increased plaque size. However, further investigation is needed to confirm and elucidate the underlying mechanisms.

Additional requirements:

1) Please ensure that your manuscript meets PLOS ONE’s style requirements, including those for file naming. The PLOS ONE style templates can be found at https://journals.plos.org/plosone/s/file?id=wjVg/PLOSOne_formatting_sample_main_body.pdf

We reformatted the manuscript to ensure alignment with PLOS ONE’s style requirements, including those for file naming.

2) In order to comply with PLOS ONE's guidelines for non-human primate experiments (http://journals.plos.org/plosone/s/submission-guidelines#loc-non-human-primates), please provide additional details regarding housing conditions, feeding regimens, environmental enrichment, and all relevant steps taken to alleviate suffering. Also indicate how often animal care staff monitored the health and well-being of the animals and the criteria used to make such assessments. Lastly, specify the disposition of animals at the end of the study (e.g. euthanasia, returned to home colony, etc.).

We added additional details on NHP housing conditions, feeding regimens, environmental enrichment etc. to the Materials and methods lines 251 – 253 and the disposition of animals at the end of the study to lines 260 and 261.

3) In your Methods section, please include a comment about the state of the animals following this research. Were they euthanized or housed for use in further research? If any animals were sacrificed by the authors, please include the method of euthanasia and describe any efforts that were undertaken to reduce animal suffering.

Information on the disposition of animals at the end of the study was added to Materials and methods lines 260 and 261.

4) We note that your Data Availability Statement is currently as follows: [All relevant data are within the manuscript and its Supporting Information files.] Please confirm at this time whether or not your submission contains all raw data required to replicate the results of your study. Authors must share the “minimal data set” for their submission. PLOS defines the minimal data set to consist of the data required to replicate all study findings reported in the article, as well as related metadata and methods (https://journals.plos.org/plosone/s/data-availability#loc-minimal-data-set-definition).

We added a Supplementary Information file (S1 File) containing the values behind the means and standard deviations used to build graphs. 

5) Please include a separate caption for each figure in your manuscript.

We inserted a separate caption for each figure into the body of the manuscript.

6) Please include captions for your Supporting Information files at the end of your manuscript, and update any in-text citations to match accordingly. Please see our Supporting Information guidelines for more information: http://journals.plos.org/plosone/s/supporting-information.

We added captions for our Supporting Information files to the end of the manuscript as per the supporting information guidelines.

We reviewed the reference list to ensure that it is complete and correct. We have not made any changes to the list, and it does not contain any retracted papers.

---

## [Decision Letter · Decision Letter 1]

27 Feb 2024

PONE-D-24-02628R1Mutations in the F protein of the live-attenuated respiratory syncytial virus vaccine candidate ∆NS2/∆1313/I1314L increase the stability of infectivity and content of prefusion F proteinPLOS ONE

Dear Dr. Zhang,

Thank you for submitting your manuscript to PLOS ONE. After careful consideration, we feel that it has merit but does not fully meet PLOS ONE’s publication criteria as it currently stands. Therefore, we invite you to submit a revised version of the manuscript that addresses the points raised during the review process.

Please respond to the reviewer's comments and highlight all edits in a revised manuscript.==============================

We look forward to receiving your revised manuscript.

Kind regards,

Ralph A. Tripp

Academic Editor

PLOS ONE

Journal Requirements:

Additional Editor Comments:

Please respond to the reviewers comments and highlight edits in a revised manuscript.

Reviewers' comments:

Reviewer's Responses to Questions

**Comments to the Author**

1. If the authors have adequately addressed your comments raised in a previous round of review and you feel that this manuscript is now acceptable for publication, you may indicate that here to bypass the “Comments to the Author” section, enter your conflict of interest statement in the “Confidential to Editor” section, and submit your "Accept" recommendation.

Reviewer #1: (No Response)

2. Is the manuscript technically sound, and do the data support the conclusions?

Reviewer #1: Yes

3. Has the statistical analysis been performed appropriately and rigorously? 

Reviewer #1: Yes

4. Have the authors made all data underlying the findings in their manuscript fully available?

Reviewer #1: Yes

5. Is the manuscript presented in an intelligible fashion and written in standard English?

Reviewer #1: Yes

6. Review Comments to the Author

Reviewer #1: In their revised and resubmitted article ”Mutations in the F protein of the live-attenuated respiratory syncytial virus vaccine candidate ΔNS2/Δ1313/I1314L increase the stability of infectivity and content of prefusion F protein" by Alamares-Sapuay et al., the authors compare two respiratory syncytial viruses. ΔNS2/Δ1313/I1314L (called ΔNS2) with a deletion of the NS2 protein and attenuating mutations in the polymerase is currently being tested in clinical studies. Into this virus four mutations were introduced into the fusion protein which have been shown to stabilize the protein in prior studies (called ΔNS2-L19F-4M). The viruses were tested for the production of prefusion-F protein and virus stability in tissues culture growth as well as growth kinetics, peak titers and antibody induction in African green monkeys. In tissue culture, prefusion F protein expression and virus stability of ΔNS2-L19F-4M was higher than that of ΔNS2. In vivo, however, no difference was found between the two viruses in respect to growth kinetics, peak titers and antibody induction after infection of African Green monkeys.

Major comments

The authors claim in their abstract that “these results suggest that ΔNS2-L19F-4M is an improved live attenuated vaccine candidate that merits further evaluation in a clinical trial in humans”. What the authors have shown is that even an increase in the expression of the prefusion/fusion protein ratio and increased virus stability in tissue culture does not result in any measurable differences in vivo. Thus, the conclusion should be that ΔNS2-L19F-4M does not promise to be superior to ΔNS2.

It is unfortunate that the authors did not challenge the immunized animals to compare immunization efficiency. If these data are available, they should be added to the manuscript.

Minor comments

Line 75: update that a second vaccine is on the market

Line 75-76: vaccination is not a passive approach, rephrase

Line 100-101 provide reference

7. PLOS authors have the option to publish the peer review history of their article (what does this mean?). If published, this will include your full peer review and any attached files.

Reviewer #1: No

---

## [Author Response · Author response to Decision Letter 1]

18 Mar 2024

Dear Editors,

I would like to extend our appreciation once more to the reviewers for their critical comments and feedback on our revised manuscript: PONE-D-24-02628 “Mutations in the F protein of the live-attenuated respiratory syncytial virus vaccine candidate ∆NS2/∆1313/I1314L increase the stability of infectivity and content of prefusion F protein”.

Regarding all new comments received, we have thoroughly addressed each point, provided comprehensive responses which can be found below and made the suggested revisions to the manuscript including incorporation of additional references. We are confident that the revised manuscript now meets all necessary criteria for publication in PLOS ONE. 

Sincerely,

Dr. Linong Zhang

Senior expert scientist, Sanofi

Responses to Reviewers' comments:

Major Comments:

• The authors claim in their abstract that “these results suggest that ΔNS2-L19F-4M is an improved live attenuated vaccine candidate that merits further evaluation in a clinical trial in humans”. What the authors have shown is that even an increase in the expression of the prefusion/fusion protein ratio and increased virus stability in tissue culture does not result in any measurable differences in vivo. Thus, the conclusion should be that ΔNS2-L19F-4M does not promise to be superior to ΔNS2.

• Answer

o In the Discussion section, we had previously explained how the increased stability of the ΔNS2-L19F-4M candidate could potentially improve the manufacturability, storage and distribution of the ΔNS2 vaccine currently in phase III clinical trials (lines 464 -476). In response to the reviewer’s comment, we have modified the abstract (Lines 47 and 48) and the concluding paragraph of the Discussion (Lines 521 – 524) to make this critical point more explicit.

• It is unfortunate that the authors did not challenge the immunized animals to compare immunization efficiency. If these data are available, they should be added to the manuscript.

• Answer

o The AGM model is well suited to investigating differences in immunogenicity and attenuation of live attenuated RSV vaccine candidates delivered intranasally. However, in our previous AGM challenge studies, we observed no distinctions among different live attenuated RSV vaccine candidates regarding protection, as all candidates provided complete protection against both upper and lower airway virus shedding in AGMs. This outcome may be linked to the semi-permissive nature of RSV infection in AGMs and the strong mucosal immunity induced by the vaccine candidates through intranasal immunization. Consequently, we deemed it unnecessary to conduct AGM challenge studies aimed at distinguishing between the live attenuated vaccine candidates. 

Minor comments:

• Line 75: update that a second vaccine is on the market. 

• Answer

o We re-wrote lines 74 – 80 to explicitly list all 3 products currently licensed to protect infants from RSV, adding a reference for Palivizumab (Ref #12).

• Line 75-76: vaccination is not a passive approach, rephrase

• Answer

o We re-wrote lines 74 – 80 to clarify that only mAb prophylaxis is a passive approach.

• Line 100-101 provide reference

• Answer

We added references for G mediated attachment via CX3C on HAE (Ref #28-30).

---

## [Editor Report · Decision Letter 2]

22 Mar 2024

Mutations in the F protein of the live-attenuated respiratory syncytial virus vaccine candidate ∆NS2/∆1313/I1314L increase the stability of infectivity and content of prefusion F protein

PONE-D-24-02628R2

Dear Dr. Zhang,

We’re pleased to inform you that your manuscript has been judged scientifically suitable for publication and will be formally accepted for publication once it meets all outstanding technical requirements.

Kind regards,

Ralph A. Tripp

Academic Editor

PLOS ONE

Additional Editor Comments (optional):

The authors have satifacorily responses to the reviewers comments.
---

## [Editor Report · Acceptance letter]

30 Mar 2024

PONE-D-24-02628R2 

PLOS ONE

Dear Dr. Zhang, 

I'm pleased to inform you that your manuscript has been deemed suitable for publication in PLOS ONE. Congratulations! Your manuscript is now being handed over to our production team.

Kind regards, 

on behalf of

Dr. Ralph A. Tripp 

Academic Editor

PLOS ONE